# Intracellular pathogen effector reprograms host gene expression by inhibiting mRNA decay

Yevgen Levdansky [1,2], Justin C. Deme [1,2], David J. Turner[1,2], Claire T. Piczak[1,2], Filip Pekovic [1], Anna L. Valkov[1], Sergey G. Tarasov [1], Susan M. Lea [1] ✉ & Eugene Valkov [1] ✉

*Legionella pneumophila*, an intracellular bacterial pathogen, injects effector proteins into host cells to manipulate cellular processes and promote its survival and proliferation. Here, we reveal a unique mechanism by which the *Legionella* effector PieF perturbs host mRNA decay by targeting the human CCR4–NOT deadenylase complex. High-resolution cryo-electron microscopy structures and biochemical analyses reveal that PieF binds with nanomolar affinity to the NOT7 and NOT8 catalytic subunits of CCR4–NOT, obstructing RNA access and displacing a catalytic $Mg^{2+}$ ion from the active site. Additionally, PieF prevents NOT7/8 from associating with their partner deadenylases NOT6/6L, inhibiting the assembly of a functional deadenylase complex. Consequently, PieF robustly blocks mRNA poly(A) tail shortening and degradation with striking potency and selectivity for NOT7/8. This inhibition of deadenylation by PieF impedes cell cycle progression in human cells, revealing a novel bacterial strategy to modulate host gene expression.

Bacterial pathogens have evolved sophisticated RNA regulatory mechanisms to sense and adapt to their environment, evade host immune defenses, and replicate. *Neisseria meningitidis* and *Streptococcus pneumoniae* use non-coding RNAs (cia-dependent small RNAs, which are a group of small RNAs that are controlled by the two-component regulatory systems, CiaRH and NrrF, respectively) to regulate key virulence factors, such as capsule formation, metal ion transport, and DNA uptake, allowing adaptation to environmental cues and iron homeostasis during infection[1,2].

Beyond regulating their genes, some bacteria manipulate host RNA machinery to enhance their survival. *Legionella pneumophila*, a Gram-negative bacterium, infects amoebae and human cells using over 300 effector proteins secreted via its Dot/Icm type IV secretion system[3]. While most effectors are genetically redundant and individually non-essential for intracellular replication, collectively, they enable *Legionella* to adapt to diverse hosts by modulating cellular processes such as vesicle trafficking, protein synthesis, and cell death[4,5].

*Legionella* infection modifies the expression of 85 human microRNAs (miRNAs), disrupting post-transcriptional regulation in macrophages and promoting bacterial replication[6]. Intriguingly, *Legionella* can translocate its small RNAs (sRNAs) into host cells to mimic miRNA functions, downregulating immune regulators and enhancing bacterial survival[7]. Recently, the effector LegK1 was reported to inhibit small interfering RNA (siRNA) and miRNA activities in human cells through its kinase function and ability to bind Argonaute proteins–critical mediators of miRNA-mediated gene silencing[8].

In addition to manipulating miRNA pathways, *Legionella* subverts host post-transcriptional control by using effectors to inhibit host protein synthesis[9–11]. When infecting macrophages, virulent *Legionella* secretes effectors that globally inhibit host protein synthesis, blocking IκB production and preventing NF-κB–mediated upregulation of pro-inflammatory cytokines[12]. Similar strategies are observed in plant pathogens; for example, the *Xanthomonas citri* effector PthA4 inhibits *Citrus sinensis* mRNA deadenylase activity to facilitate virulence[13].

[1]National Cancer Institute, National Institutes of Health, Frederick, MD, USA. [2]These authors contributed equally: Yevgen Levdansky, Justin C. Deme, David J. Turner, Claire T. Piczak. ✉e-mail: susan.lea@nih.gov; eugene.valkov@nih.gov

Bacterial pathogens intricately manipulate host RNA regulatory systems, including targeting mRNA decay and silencing machinery, to facilitate infection and promote their survival. PieF, an effector encoded in a highly dynamic genomic locus of *Legionella pneumophila* that also encodes other effectors[14], was recently shown to interact with the conserved multi-subunit CCR4-NOT complex[15,16]. This complex is essential in eukaryotic cells for initiating mRNA decay through the enzymatic shortening (deadenylation) of mRNA poly(A) tails[17]. Here, we reveal the potent and selective mechanism by which PieF targets the host CCR4-NOT complex: by binding with high affinity to its deadenylase subunits, PieF hinders the assembly of a deadenylation-competent CCR4-NOT complex. Overexpression of PieF in human cells induces cell cycle arrest, suggesting that post-infection targeting of host mRNA deadenylation machinery may serve as a strategy to assist with pathogen survival and replication.

## Results

### Legionella pneumophila PieF effector binds to specific CCR4-NOT subunits

The CCR4-NOT complex contains two deadenylases, NOT7/8 and NOT6/6L, belonging to the DEDD and EEP-type exonuclease families, respectively[17] (Fig. 1a). To validate the reported PieF interaction with NOT7/8 and assess potential binding to other CCR4-NOT subunits, we performed a direct interaction assay by pull-down with purified recombinant proteins. N-terminally StrepII-tagged PieF efficiently bound NOT7 and NOT8 with apparent stoichiometry (Fig. 1b). However, NOT9, a non-enzymatic subunit of the CCR4-NOT positioned close to the deadenylases within the complex, did not bind PieF under the same conditions. As a positive control, a fragment of RNF219, a known interactor of the NOT9 subunit, efficiently bound to NOT9. (Fig. 1b)[18]. Additionally, NOT8, the NOT7 paralog, bound PieF (Fig. 1b).

NOT7 is a component of the CCR4-NOT catalytic module[17,19,20]. Beyond its catalytic function, NOT7 connects the endonuclease-exonuclease-phosphatase (EEP) type deadenylase NOT6 (or its paralog NOT6L) to the central MIF4G domain of NOT1 (Fig. 1a)[19,21,22]. We investigated the potential interaction between PieF and NOT7 within the NOT7:NOT6 and NOT7:NOT6L heterodimer contexts. Pull-down assays showed no detectable interaction between PieF and these heterodimers, indicating that NOT6/6L binding likely precludes PieF association (Fig. 1c). However, a systematic quantitative analysis of the ratios of CCR4-NOT subunits in human cells using mass spectrometry revealed that NOT6 is strikingly substoichiometric compared to other CCR4-NOT subunits[23]. These data suggest that the predominant form of CCR4-NOT in human cells may be devoid of NOT6/6L subunits leaving NOT7/8 subunits available for potential interaction with PieF.

### PieF is a potent and specific inhibitor of NOT7/8 deadenylation activity

We then investigated PieF's impact on NOT7's deadenylation activity in isolation and within the NOT6:NOT7 heterodimer. Using an in vitro deadenylation assay, we tested NOT7 and NOT6:NOT7 with and without an equimolar amount of PieF (Fig. 1d,e, Supplementary Fig. 1a,b). The substrate was a 5′-fluorescently-labeled UCUACAU-A$_{20}$ synthetic RNA.

While the NOT6:NOT7 heterodimer's activity remained unchanged, NOT7 alone was inhibited ~5-fold by PieF. In the same experimental setup, NOT8 (which has ~3-fold lower basal activity than NOT7) was inhibited ~6-fold by PieF (Supplementary Fig. 1c,d).

Various DEDD-type exoribonucleases are present in cells, including those capable of poly(A) tail shortening, such as poly(A)-specific ribonuclease (PARN)[24]. We purified recombinant PARN and observed that PARN is not inhibited by PieF, unlike NOT7/8. This suggests that PieF specifically targets the deadenylation activity of the CCR4-NOT complex rather than affecting DEDD-type exoribonucleases in general (Supplementary Fig. 1e).

We used isothermal titration calorimetry (ITC) to determine PieF's binding affinity for NOT7/8. The measured dissociation constant ($K_d$) values for NOT7 and NOT8 were ~24 nM, indicating high-affinity binding (Fig. 1f and Supplementary Fig. 1f; Supplementary Table 1).

Given the mutual exclusivity of NOT6 and PieF binding to NOT7 and PieF's low-nanomolar range affinity, we investigated whether PieF could outcompete NOT6 from the NOT6:NOT7 heterodimer. We purified and immobilized NOT6:NOT7 on beads, then titrated PieF up to 200-fold molar excess. Surprisingly, we did not observe the formation of the NOT7:PieF complex or displacement of NOT6 (Supplementary Fig. 1g).

We employed analytical size exclusion chromatography to test whether PieF interferes with the NOT7 and NOT1 association. We found that NOT7 simultaneously binds the NOT1 MIF4G domain and PieF, forming a NOT1:NOT7:PieF ternary complex (Fig. 2a). Mass photometry analysis revealed a stable 1:1:1 complex within a single symmetrical peak (Fig. 2b).

### Cryo-EM reveals how PieF mimics NOT6/6 L and inhibits NOT7/8

To understand the binding dynamics of NOT7 with NOT1, PieF, and NOT6 and to elucidate the mechanism of PieF-mediated NOT7/8 inhibition, we determined the structure of the NOT1:NOT7:PieF complex to 2.8 Å resolution and NOT1:NOT8:PieF complex to 3.5 Å resolution using cryo-electron microscopy (cryo-EM) (Fig. 2c, d; Supplementary Table 2). The high quality of the density map enabled us to build a complete NOT1 MIF4G domain (residues E1093–S1317). NOT7 contains most residues (Q10–G263 out of 285), with only a few flexible residues missing at both termini, while PieF includes all residues except for I19–V25 in the loop connecting the β2 and β3 strands (Fig. 2e, f and Supplementary Fig. 2). Using Foldseek analysis[25], we determined that the PieF of *Legionella pneumophila* has a fold, consisting of a five-stranded beta sheet and two alpha helices, with no close homologs in PDB, CATH50, AFDB-Proteome, and AFDB-SwissProt databases.

PieF and NOT7 form a complex through a bipartite interface with two distinct interaction regions (Fig. 2e, f): the first interface involves PieF's β5 strand and adjacent loops, while the α3 helix of PieF mediates the second interface.

A striking feature of the latter interface is how PieF binds NOT7 using its α3 helix. This interaction occludes NOT7's active site and involves PieF's K124 displacing one of the two catalytic Mg$^{2+}$ ions. This displacement occurs due to the formation of salt bridges between the side chains of K124$^{PieF}$ with D40$^{NOT7}$ and D161$^{NOT7}$ (Fig. 3a, b). K124$^{PieF}$ further hydrogen bonds with the carbonyl group of F156$^{NOT7}$, and via its carbonyl group, K124$^{PieF}$ interacts with H157$^{NOT7}$ (Supplementary Fig. 3a). The α3 helix also makes stabilizing hydrophobic contacts: L121$^{PieF}$ interacts with Y160$^{NOT7}$ and F43$^{NOT7}$, and Y119$^{PieF}$ interacts with L115$^{NOT7}$ (Supplementary Fig. 3a).

The principal interface between PieF and NOT7 is primarily formed by a section of PieF's β5 strand (residues S101–F103) and several adjacent residues (K96–G100) immediately preceding it (Supplementary Fig. 3b). The β5 strand of PieF assembles into a short β-sheet with the β3 strand of the NOT7 (residues A48–P50). Additionally, R49$^{NOT7}$ forms a salt bridge with E117$^{PieF}$ (Supplementary Fig. 3b). In addition to creating a β-sheet, F103$^{PieF}$ along with I88$^{PieF}$, K96$^{PieF}$, and I97$^{PieF}$, establish a hydrophobic network of contacts converging on with L71$^{NOT7}$. The loop between PieF's β5 strand and α3 helix (residues F105–E114) also interacts with NOT7, stabilizing the α3 helix. F111$^{PieF}$ makes another set of critical hydrophobic contacts with V47$^{NOT7}$, while its carbonyl group interacts with R113$^{PieF}$, stabilizing both F111$^{PieF}$ and S110$^{PieF}$.

The strong interactions between NOT6 and NOT7, forming a stable complex with an effectively negligible off-rate, account for PieF's inability to displace NOT6 from the heterodimer. Specifically, NOT6's β2 strand (residues E33–I36) forms a β-sheet with NOT7's

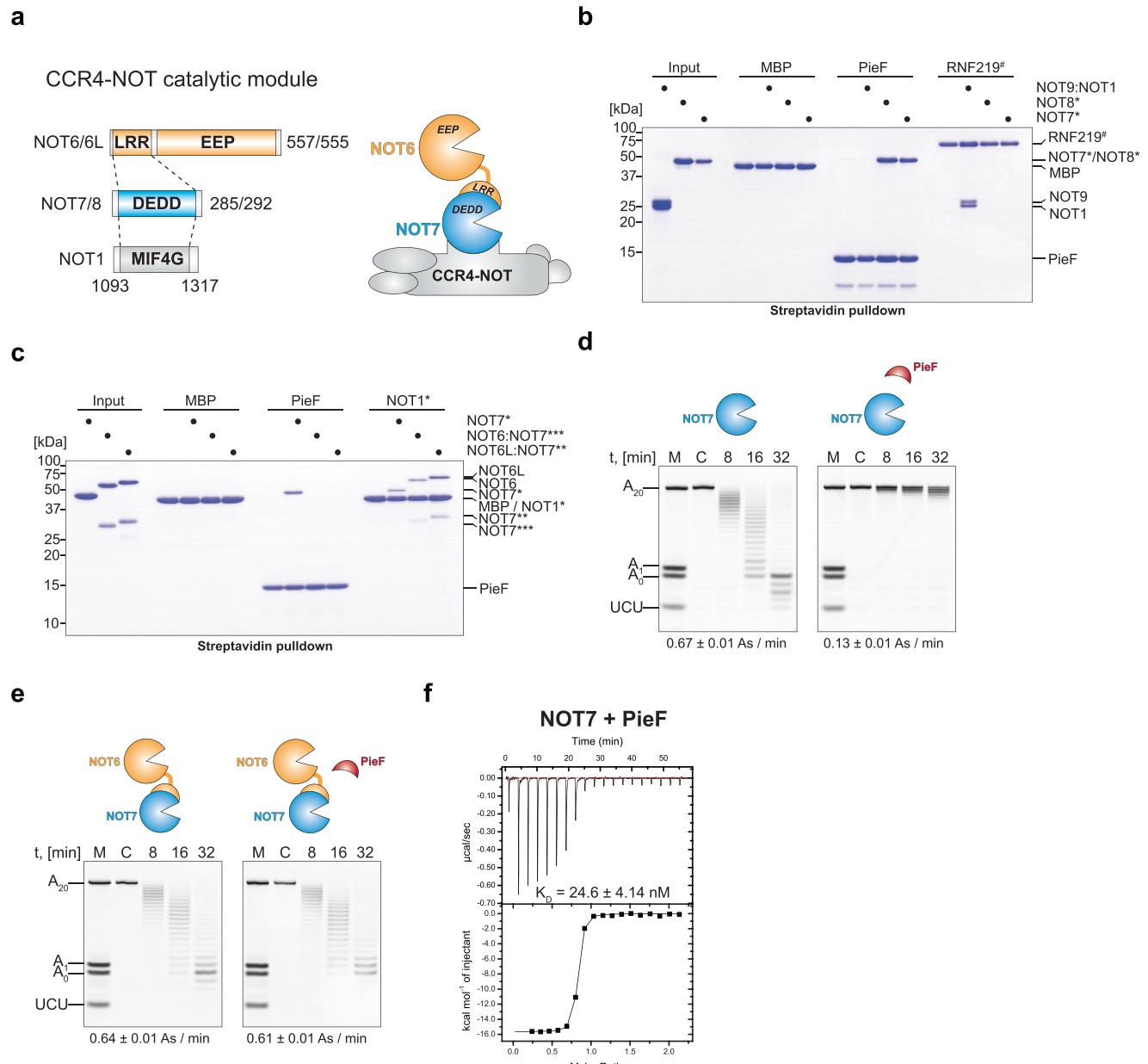

**Fig. 1 | *Legionella* effector PieF binds NOT7 and NOT8 deadenylase enzymes.**
**a** Domain organization and schematic representation of the catalytic module of the human CCR4-NOT complex. **b** Coomassie-stained 15% polyacrylamide gel of the in vitro streptavidin pull-down assay with recombinant N-terminally StrepII-tagged PieF upon incubation with NOT9 module and His$_6$-SUMO-tagged NOT7 and NOT8 deadenylases. * denotes the presence of an N-terminal His$_6$-SUMO-tag. # denotes an N-terminal His$_6$-MBP-tag. Experiment was independently repeated multiple times ($n > 3$). **c** Coomassie-stained 15% polyacrylamide gel of the in vitro streptavidin pull-down assay with recombinant N-terminally StrepII-tagged PieF upon incubation with His$_6$-SUMO-tagged NOT7*, NOT6:NOT7*** and NOT6L:NOT7** heterodimers. * denotes presence of an N-terminal His$_6$-SUMO-tag, ** denotes an N-terminal His$_8$-tag. *** denotes untagged NOT7. Experiment was independently repeated multiple times ($n > 3$). **d, e** In vitro deadenylation assays with 50 nM of UCUACAU-A$_{20}$ RNA substrate, His$_6$-SUMO-NOT7 (500 nM; **d**) and NOT6:NOT7 heterodimer (250 nM; **e**)

without (left panels) and with (right panels) the equimolar to corresponding deadenylase amount of N-terminally His$_6$-tagged PieF. M indicates the tail length RNA marker, and C represents the control sample without the deadenylase. Poly (A) tail length changes were quantified by plotting the most abundant tail length at each time point. Linear regression was used to determine the apparent deadenylation rate (As/min); values are presented as mean $\pm$ SE ($n = 3$). The molecular size markers of right panels are identical to the ones on left. **f** Representative isothermal titration calorimetry (ITC) thermograms of the interaction between His$_6$-tagged PieF and His$_6$-SUMO-NOT7. The upper panel shows raw data in (μcal s$^{-1}$), and the lower panel represents the integration of heat changes associated with each injection (kcal mol$^{-1}$ of injectant). Data were fitted using a one-site binding model. The parameters of the runs are summarized in Supplementary Table 1. The schematics in (**a**, **d**, **e**) were drawn using Adobe Illustrator 2025. Source data are provided as a Source Data file.

β3 strand (residues A48–I51)[22]. This interaction interface directly overlaps with PieF's binding site on NOT7 (Fig. 3c, d, Supplementary Fig. 3c, d), underscoring the mutual exclusivity of these protein interactions.

Within the NOT6:NOT7 heterodimer, R49$^{NOT7}$ plays a crucial role, forming a salt bridge with E35$^{NOT6}$ and hydrogen bonding with the

carbonyl of R12$^{NOT6}$ (Supplementary Fig. 3c, d). Despite this relatively small overlapping interface, PieF surprisingly cannot displace NOT6 from the NOT6:NOT7 heterodimer. We hypothesized that inability to displace NOT6/6L may be due to an additional binding interface, where the N-terminal loop of NOT6/6L wraps around NOT7's α2 helix and adjacent loop between the helix and β3 strand (Fig. 3c, d,

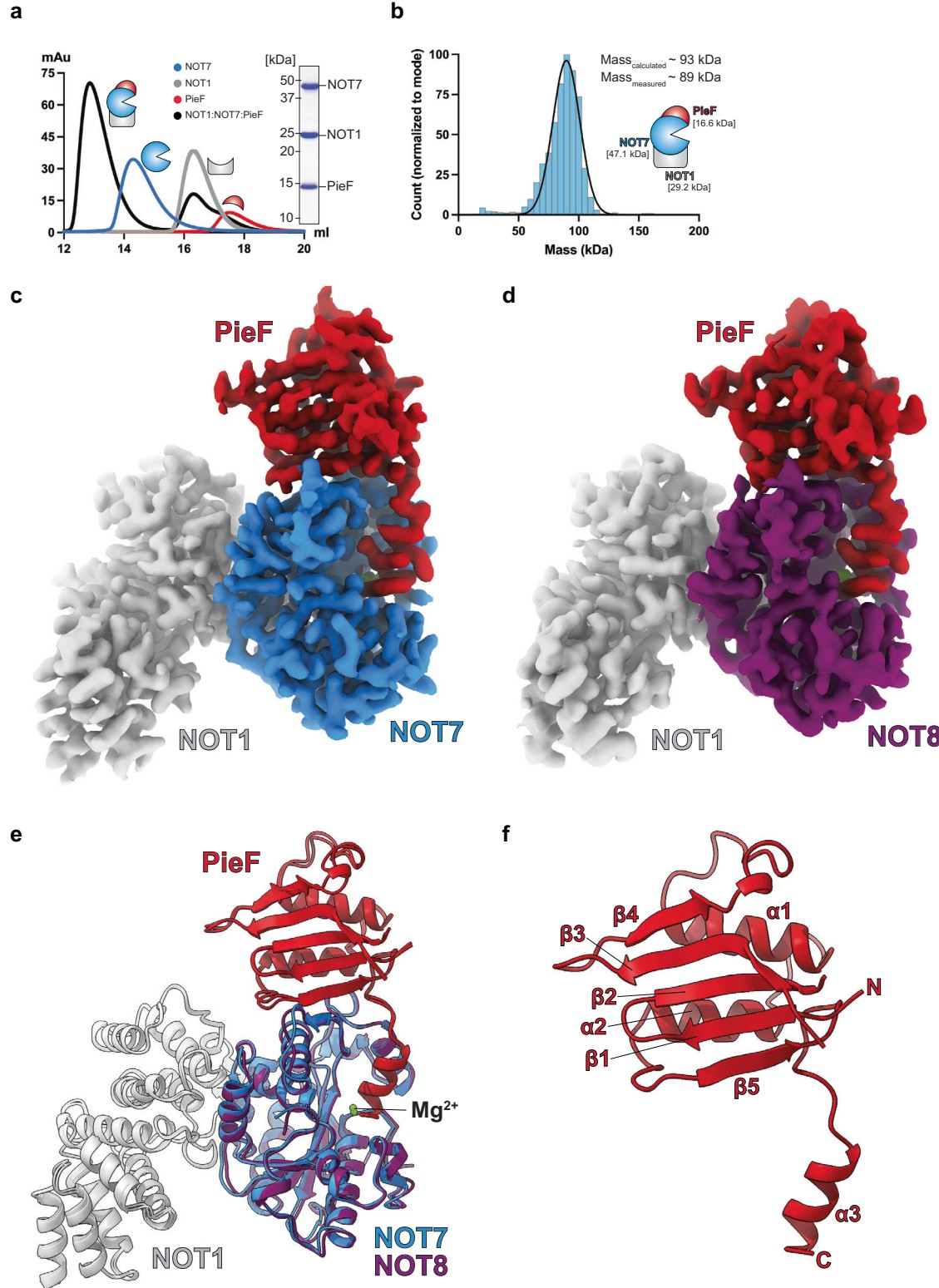

**Fig. 2 | Reconstitution and cryo-EM structures of the NOT1:NOT7/8:PieF ternary complexes. a** Analytical size exclusion chromatography profiles (left panel) and a 15% Coomassie-stained gel (right panel) of the purified components and intermediates of the NOT1:NOT7:PieF complex. The tags used to purify the individual components were preserved during the reconstitution of the complex. Experiment was independently repeated multiple times (*n* > 3). Source data are provided as a Source Data file. **b** Mass photometry analysis of the NOT1:NOT7:PieF complex. The tags used for purification were preserved during the reconstitution of the complex. **c** Cryo-EM reconstruction of the NOT1:NOT7:PieF complex at 2.8 Å resolution. **d** Cryo-EM reconstruction of the NOT1:NOT8:PieF complex at 3.5 Å resolution. **e** Superimposed structures of the NOT1:NOT7/8:PieF complexes in cartoon representation. **f** Secondary structure annotation of PieF structure. The schematics in (**a**, **b**) were drawn using Adobe Illustrator 2025.

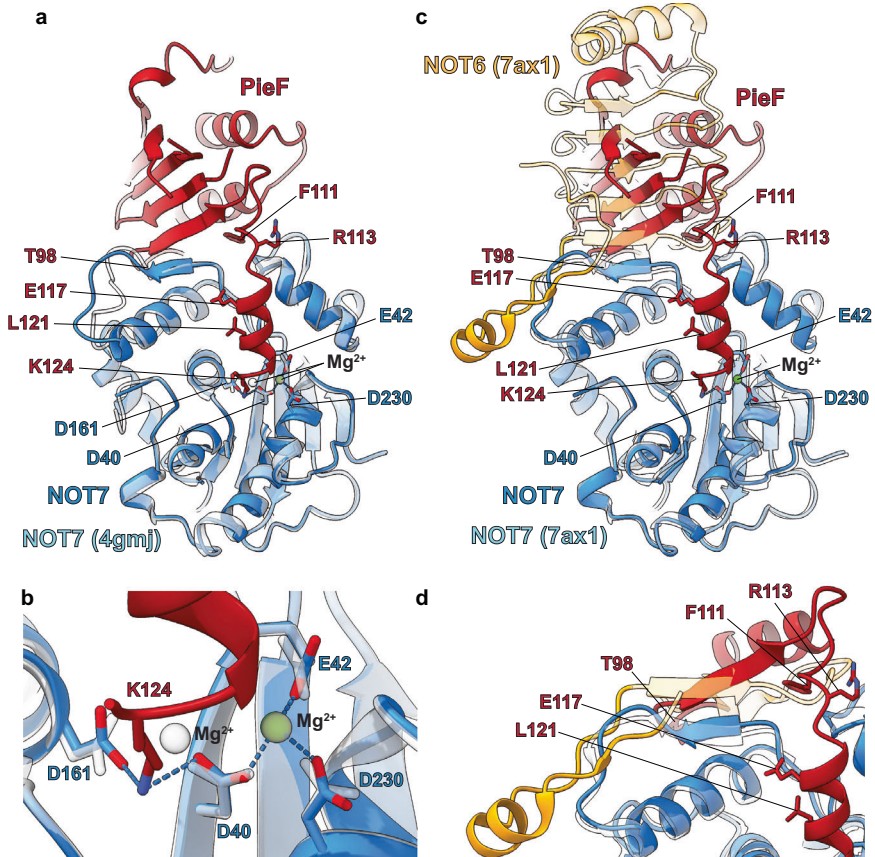

**Fig. 3 | Structural details of the NOT7:PieF interfaces. a** Cartoon representation of the NOT7 (blue) active site and part of the helix α3 of PieF (red). Residues K206–E211 were omitted for clarity. The residues involved in the interaction are represented as sticks and the Mg²⁺ atom is shown as a green sphere. Dashed lines indicate metal coordination (NOT7 and Mg²⁺) or hydrogen bonding (NOT7 and PieF). D40$^{NOT7}$ and D161$^{NOT7}$ are additionally involved in salt bridge interactions with K124$^{PieF}$. Interaction of K124$^{PieF}$ with D40$^{NOT7}$ and D161$^{NOT7}$ displaces a catalytic Mg²⁺ from the active site, and the NOT7 crystal structure (PDB: 4gmj; transparent blue) is superimposed to illustrate this. **b** Close-up view of (**a**). **c** NOT7:PieF, as represented in (**a**), superimposed on the crystal structure of the NOT6:NOT7 heterodimer (PDB: 7ax1). The leucine-rich repeat region of NOT6 binds to the same site of NOT7 as PieF. The N-terminal region of NOT6 (solid yellow) extends the binding interface. **d** Close-up view of (**c**).

Supplementary Fig. 3c–f, Supplementary Fig. 4a). To test the hypothesis, we purified the version of the NOT6:NOT7 heterodimer, in which NOT6 lacks residues M1–K28, and did the pull-down assays in presence of the excess of PieF. Even with the truncated N-terminal portion of NOT6, PieF was still unable to outcompete NOT6 from the heterodimer due to the continued stability of the NOT6:NOT7 interface, which likely exhibits very slow dissociation kinetics (Supplementary Fig. 4a–c). In deadenylation assays, we observed no inhibition of the NOT6$_{Δ1-28}$:NOT7 heterodimer activity by PieF further corroborating the pull-down results (Supplementary Fig. 4d).

We determined the cryo-EM structure of the NOT1:NOT8:PieF ternary complex at 3.5 Å resolution to further elucidate paralog-specific differences in PieF-mediated deadenylase inhibition. The resulting model includes the same NOT1 and PieF residues as in the NOT7 complex, while NOT8 is nearly complete, missing only the first five and last 29 residues (V6–G263 out of 292) (Fig. 2e, Supplementary Fig. 2). NOT8 associates with NOT1 in a manner nearly identical to NOT7 closely matching previously determined structures (Supplementary Fig. 4e, f)[21], suggesting that PieF does not prevent the incorporation of the inhibited deadenylase into the CCR4-NOT complex (Supplementary Fig. 4e, f). The structures of the NOT7 and NOT8 are nearly identical (Supplementary Fig. 4g, h vs 3a,b; r.m.s.d ~0.56 Å over 254 Cα pairs), with a few notable differences. NOT8 does not form a β3 strand; however, R49$^{NOT8}$ forms a salt bridge with E117$^{PieF}$, as in the NOT7 structure (Supplementary Fig. 2; Supplementary Figs. 4h vs 3b). NOT8 residues do not contact S99$^{PieF}$ and G100$^{PieF}$. At the same time,

T98$^{PieF}$ interacts with the R173$^{NOT8}$ (N173$^{NOT7}$), which additionally interacts with the carbonyl group of K96$^{PieF}$, a contact absent in the NOT7 structure (Supplementary Figs. 4h vs 3b). Although highly similar, NOT7 and NOT8 exhibit their main structural and sequence divergence beyond residue 265—a region likely flexible and unresolved in either cryo-EM structure (Supplementary Fig. 5).

## The C-terminal helix enhances PieF inhibition through synergistic binding

To elucidate the roles of specific PieF residues in their interaction with NOT7 and subsequent inhibition, we employed a structure-guided approach to generate several PieF mutants (Fig. 4a). We then evaluated these mutants through deadenylation assays with NOT7 to assess inhibitory effects and isothermal titration calorimetry (ITC) to measure changes in binding affinity.

We first examined PieF$^{K124A}$ to assess if removing salt bridges with D40$^{NOT7}$ and D161$^{NOT7}$ would fully relieve inhibition and tested PieF$^{K124R}$ for functional substitution. Under identical conditions to wild-type PieF, deadenylation assays showed NOT7's activity with PieF$^{K124A}$ was ~2.5-fold faster than with wild-type PieF, while ITC revealed ~23-fold weaker binding (Fig. 4b, Supplementary Fig. 6b, c). PieF$^{K124R}$ had milder effects: ~1.2-fold faster deadenylation and ~4-fold decreased affinity (Fig. 4b, Supplementary Fig. 6b, d). These results indicate salt bridge interactions with catalytic residues are crucial for PieF's total inhibitory capacity.

To further investigate the residues crucial for PieF's interaction with and inhibition of NOT7, we created a PieF 5M mutant (Fig. 4a;

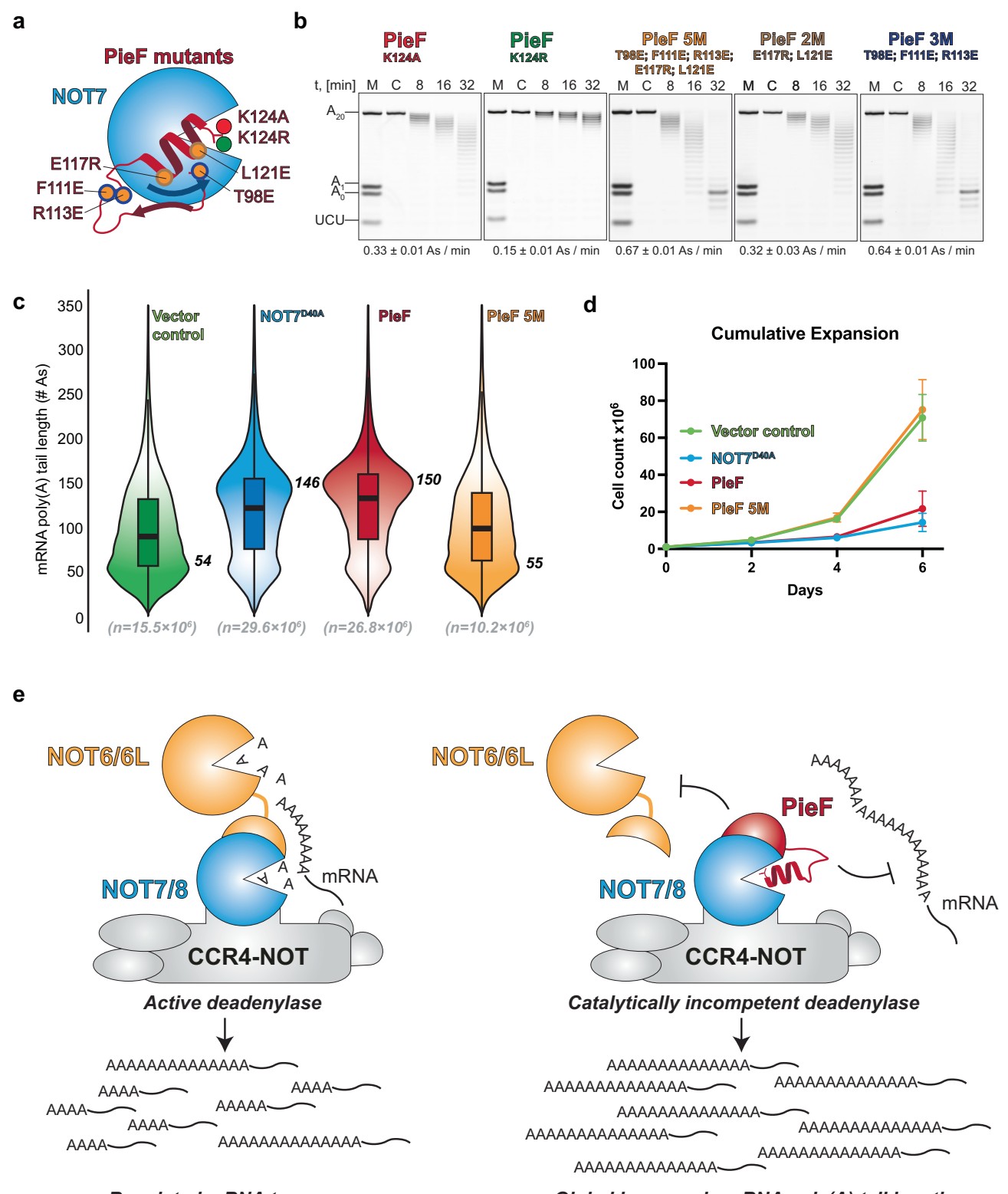

behaved similarly to PieF$^{K124A}$, with ~2.5-fold faster deadenylation and ~18-fold weaker binding than wild-type PieF (Fig. 4b, Supplementary Fig. 6b, f). PieF 3 M mirrored PieF 5M, showing no inhibition or binding to NOT7 (Fig. 4b, Supplementary Fig. 6b, g). These results suggest that, although the PieF C-terminal helix contributes to inhibition, full inhibitory function depends on stabilization and proper orientation by proximal residues.

orange circles): two mutations in the helix (E117R and L121E; 2M, brown rim) and three around the β5 strand (T98E, F111E, and R113E; 3M, dark blue rim). PieF 5M showed no inhibition of NOT7 deadenylation or detectable binding (Fig. 4b, Supplementary Fig. 6b, e). We then wondered which mutations were responsible for the loss of PieF-mediated inhibition: those in the helix (2M) or around the β5 strand (3M). To address this, we tested the 2M and 3M mutants separately. PieF 2M

**Fig. 4 | Validation of the PieF inhibitory interface in deadenylation assays and in cells. a** Schematic representation of the mutated residues in PieF used in this study. **b** In vitro deadenylation assays with 50 nM of UCUACAU-A$_{20}$ RNA substrate, 500 nM of His$_6$-SUMO-NOT7 and 500 nM of His$_6$-tagged PieF$^{K124A}$, PieF$^{K124R}$, PieF 5 M, PieF 2 M, and PieF 3 M. The molecular size markers on all gels are identical to those shown in the leftmost panel. Experiment was independently repeated multiple times ($n = 3$). Source data are provided as a Source Data file. **c** Violin plots showing the poly(A) tail distribution in HEK293T cells expressing transfected GFP control (green), GFP-NOT7$^{D40A}$ (blue), GFP-PieF (dark red), and GFP-PieF 5M (orange) based on mRNA poly(A) length estimates by nanopore sequencing. Modal poly(A) tail length values are next to the corresponding violin plot. Box plots show median

value (solid bold line) and 1st and 3rd quartiles, and whiskers represent 1.5 × IQR (interquartile range). *n* is the number of basecalled reads in each sample. **d** Cumulative cell counts at day 2, day 4, and day 6 after transfection and over-expression of a control GFP construct (green) compared to GFP-NOT7$^{D40A}$ (blue), GFP-PieF WT (dark red) or GFP-PieF 5M (orange) constructs. Values are presented as mean ± standard deviation ($n = 3$). Source data are provided as a Source Data file. **e** The *Legionella pneumophila* effector PieF or NOT6 bind to NOT7 within the CCR4-NOT complex in a mutually exclusive manner. PieF binding impairs the assembly and deadenylation activity of the CCR4-NOT complex, leading to longer poly(A) tails, increased mRNA levels, and defects in cell cycle progression. The schematics were drawn using Adobe Illustrator 2025.

## PieF disrupts mRNA poly(A) tail control and impairs cell cycle progression in human cells

To assess PieF's impact on deadenylation in human cells, we expressed GFP-tagged PieF in HEK293T cells and measured transcriptome-wide poly(A) tail lengths using nanopore direct RNA sequencing. Despite PieF's lack of effect on preassembled NOT6/6L:NOT7/8 heterodimers in vitro, GFP-PieF expression in cells dramatically increased modal mRNA poly(A) tail length from ~54 to ~150 adenosines, consistent with a highly pronounced CCR4-NOT-mediated deadenylation impairment (Fig. 4c, green vs red). In vitro assays confirmed that GFP-PieF inhibits NOT7 as effectively as untagged PieF, corroborating the cellular data (Supplementary Fig. 6h).

To confirm that the observed deadenylation impairment resulted from CCR4-NOT dysfunction, we overexpressed GFP-tagged NOT7$^{D40A}$, a well-characterized dominant-negative mutant[26]. The modal poly(A) tail length with GFP-NOT7$^{D40A}$ (~146 adenosines) closely resembled that observed with GFP-PieF (~150 adenosines), demonstrating that PieF overexpression phenocopies NOT7 inactivation (Fig. 4c, blue). Crucially, cells overexpressing the GFP-PieF 5 M mutant exhibited mRNA poly(A) tail lengths similar to GFP alone (~55 vs. ~54 adenosines), consistent with the PieF inhibitory effect being critically dependent on its stable binding to NOT7 (Fig. 4c, orange).

Control of mRNA poly(A) tail length plays a crucial role in cell cycle progression regulation[27]. Since poly(A) tail length is primarily controlled by CCR4-NOT deadenylation activity, we investigated how PieF affects cell cycle progression by inhibiting this activity. We observed that cells expressing GFP-tagged PieF and NOT7$^{D40A}$ proliferated more slowly than those expressing GFP or GFP-PieF 5 M controls. A minor decrease in viability (from 95 to 85%) upon GFP-PieF and GFP-NOT7$^{D40A}$ overexpression only partially explains these observations (Fig. 4d and Supplementary Fig. 7a).

To assess PieF's impact on cell cycle progression via CCR4-NOT inhibition, we used DAPI staining and flow cytometry to quantify DNA content across cell cycle phases (G$_0$/G$_1$, S, G$_2$, M). This method allowed us to evaluate how PieF and its variants affected cell cycle distribution. Cells expressing GFP-PieF showed a ~10% decrease in G$_0$/G$_1$ phase frequency and a comparable increase in S phase frequency compared to GFP-expressing cells (Supplementary Fig. 7b, c, red vs green). GFP-NOT7$^{D40A}$-expressing cells exhibited a similar reduction in the G$_0$/G$_1$ phase and an increase in S phase frequencies relative to GFP vector control, consistent with a critical role of regulated mRNA decay and translational repression in maintaining normal cell cycle progression (Supplementary Figs. 7b, c, blue vs green). Notably, GFP-PieF 5 M control cells reversed this trend, emphasizing that PieF's binding to NOT7 is essential for modulating the cell cycle (Supplementary Figs. 7b, c, orange vs red). These results suggest that *L. pneumophila* employs PieF to inhibit CCR4-NOT-mediated mRNA deadenylation as an additional strategy to disrupt cell cycle signaling cascades and control the host cell cycle.

## Discussion

We show that PieF manipulates host mRNA decay by inhibiting the NOT7/8 deadenylase paralogs of the CCR4-NOT complex. This

inhibition is achieved through physical occlusion of the NOT7/8 active site and displacement of a catalytic Mg$^{2+}$ ion, combined with the prevention of native deadenylation complex assembly by blocking NOT6/6L binding to NOT7/8. The CCR4-NOT complex assembles via interactions among its subunits, with NOT6/6L binding to NOT7/8—scaffolded by the NOT1 subunit—being crucial for forming a functional deadenylase complex. Quantitative proteomic analyses[28] show that NOT6 is present at significantly lower levels (~700 copies per cell) compared to other subunits such as NOT7 (~15,000 copies per cell), NOT1 (~18,000 copies per cell), and NOT9 (~19,000 copies per cell), suggesting that NOT6 availability may indeed be limiting relative to other complex subunits[23].

Since PieF cannot outcompete NOT6 once it is bound to NOT7, its inhibitory effect is likely exerted under cellular conditions where there is a strong bias in the distribution of NOT6/6L and NOT7/8 subunits, favoring the latter. By sequestering the exposed NOT7/8 subunits in partially assembled CCR4-NOT, PieF may prevent their association with the limiting amounts of NOT6/6L. It is particularly intriguing that PieF binds to NOT7/8 in such a way that NOT7/8 can still dock onto the MIF4G domain of NOT1, thereby trapping the entire CCR4-NOT complex in a catalytically incompetent state (Fig. 4e). In this PieF-bound state, CCR4-NOT may function in a dominant-negative manner, impairing its critical roles in mRNA stability regulation and expression. This interference underscores the importance of the assembly dynamics and turnover of the CCR4-NOT complex in regulating mRNA decay. When PieF is expressed in human cells, it leads to striking transcriptome-wide perturbations in polyadenylation, including poly(A) tail lengthening, reduced mRNA decay, and consequent mRNA accumulation. The downstream effects of this disruption of mRNA homeostasis include impaired cell proliferation and delayed cell cycle S phase progression.

Our biochemical, structural, and cellular evidence suggests that *L. pneumophila* PieF has evolved to target CCR4-NOT-mediated dead-enylation by structurally mimicking the NOT7/8-interacting region of the leucine-rich repeat domain of NOT6/6L. Notably, the K124 residue, which plays a pivotal role in PieF's inhibitory properties, is unique among *Legionella* species (Supplementary Figs. 8a, 9 and 10). Interestingly, the inhibitory strategy of PieF parallels that of certain bacterial toxin-antitoxin systems[29,30]. In these complexes, the antitoxin protein neutralizes its RNase toxin by physically occluding the active site and displacing the catalytic Mg$^{2+}$ ion with a positively charged side chain. For example, the FitA antitoxin inserts an Arg residue into the FitB toxin's active site to expel the Mg$^{2+}$ required for catalysis[31]. Similarly, the VapB antitoxin uses an Arg (R79) to eject the Mg$^{2+}$ cofactor from the VapC toxin[32]. These parallels underscore a convergent inhibitory strategy among bacterial pathogens, whereby PieF, similar to bacterial antitoxins, inactivates NOT7's enzymatic core by displacing its catalytic metal ion.

Although sequence homologs of PieF are found in other bacteria, the absence of this critical K124 residue (Supplementary Figs. 8a, 9 and 10) suggests that these homologs may represent examples of a conserved effector structure being used for different biological roles. As is typical with many effectors, redundancy in these systems

complicates the identification of the specific advantage conferred by PieF's subtle disruption of mRNA homeostasis, mainly since the low levels of PieF produced during natural infection and deletion of the *pie* operon do not prevent *L. pneumophila* infections in model systems[14].

*L. pneumophila* replicates in *Acanthamoeba castellanii* and numerous other protist hosts, where impairment of host proliferation is essential for successful replication of the bacterium[33,34]. Although direct manipulation of cell cycle progression by targeting cyclins and cyclin-dependent kinases is a common feature of intracellular pathogens[34,35], the signaling cascade components can vary significantly across species, making this approach less effective across diverse hosts. In contrast, the NOT7/8 complex is highly conserved across a wide range of *L. pneumophila* hosts—from protists to higher eukaryotes (Supplementary Figs. 5, 8b and 11)—establishing it as an effective target for mRNA homeostasis manipulation with a single effector sequence. By injecting PieF, *L. pneumophila* gains the means to manipulate mRNA levels directly and the expression of genes involved in cell cycle progression, adding another regulatory mechanism to its control over host metabolism and life cycle. Thus, while PieF is not essential for infection, its ability to target NOT7/8 may confer an evolutionary advantage by expanding the pathogen's range of potential hosts.

In conclusion, we propose that PieF's unique inhibitory interface could serve as a template for developing potent, highly specific deadenylase inhibitors, given the crucial roles of the CCR4-NOT complex in various biological and disease processes ranging from T cell quiescence maintenance[36] to metastatic progression[37].

## Methods
### DNA constructs
The DNA constructs are summarized in Supplementary Table 3.

### Protein production and purification
His$_6$-PieF was produced in *E. coli* BL21(DE3) Star cells using 1 L of autoinduction media at 37 °C overnight. The cells were harvested and resuspended in a lysis buffer containing 50 mM potassium phosphate pH 7.5, 300 mM NaCl, 25 mM imidazole, and lysed by sonication. The lysate was clarified by centrifugation at 40,000 × *g* for 45 min and loaded on a 1 ml nickel-charged IMAC column (Cytiva). The bound protein was washed first with 20 column volumes of lysis buffer and then with five column volumes of lysis buffer supplemented with 35 mM imidazole. PieF was eluted from the column in 0.5 ml fractions with the same buffer supplemented with 250 mM imidazole. Peak fractions from nickel affinity chromatography were then loaded and eluted on Superdex 75 26/600 (Cytiva) equilibrated in a buffer containing 10 mM HEPES/NaOH pH 7.5, 200 mM NaCl, 2 mM DTT. The peak fractions were then pooled together, concentrated to 6–12 mg/ml, flash-frozen in liquid nitrogen, and stored at −80 °C.

All other His$_6$-PieF constructs, His$_6$-NOT1(E1093–S1317), His$_6$-SUMO-NOT7, and His$_6$-Strep-SUMO-NOT7 were produced and purified identically to the wild-type PieF.

His$_6$-GFP-PieF and His$_6$-SUMO-NOT8 were produced in *E. coli* BL21(DE3) Star cells using 2 L of autoinduction media at 20 °C overnight. His$_6$-GFP-PieF and His$_6$-SUMO-NOT8 were purified identically to the wild-type PieF.

Purifications of the NOT6:NOT7 heterodimer and NOT9 module were described previously[19,38]. His$_6$-NOT6:StrepII-SUMO-NOT7 was purified identically to NOT6:NOT7, with the only difference being that tags were not cleaved. NOT6$_{\Delta1\text{-}28}$:Strep-NOT7 was purified as described previously[22].

NOT6L:His$_8$-NOT7 was produced in *Spodoptera frugiperda* Sf21 insect cells using the MultiBac baculovirus expression system. The Sf21 cells were grown to a density of $2 \times 10^6$ cells/ml at 27 °C in Sf900II medium (Thermo Fisher Scientific), infected with the V$_1$ NOT6L:His$_8$-NOT7 stock of baculovirus, and harvested 48 h after they stopped

dividing. Cells were resuspended in lysis buffer (50 mM HEPES/NaOH pH 7.5, 500 mM NaCl) and lysed by sonication. The lysate was cleared by centrifugation at 40,000 × *g* for 1 h at 4 °C, filtered through 0.45 μm syringe-driven filters (Millipore), and loaded on a 5 ml nickel-charged IMAC column (Cytiva) equilibrated in buffer containing 50 mM potassium phosphate pH 7.5, 300 mM NaCl and 25 mM imidazole. After several washing steps, the NOT6L:His$_8$-NOT7 heterodimer was eluted with the same buffer supplemented with 250 mM imidazole. The top 5 ml were loaded and eluted on Superdex 200 16/600 equilibrated in 10 mM HEPES/NaOH pH 7.5; 200 mM NaCl; 2 mM DTT. The peak fractions were then pooled, concentrated to 7 mg/ml, flash-frozen in liquid nitrogen, and stored at −80 °C.

The expression plasmid of human PARN was a kind gift from Perry Blackshear. His$_6$-PARN was produced in *E. coli* BL21 (DE3) Star cells using 1 L of autoinduction media at 20 °C overnight. The cells were harvested and resuspended in a lysis buffer containing 50 mM HEPES/NaOH pH 7.5, 1000 mM NaCl, 5% (v/v) glycerol, 25 mM imidazole, and lysed by sonication. The lysate was clarified by centrifugation at 40,000 × *g* for 40 min and loaded on a 5 ml nickel-charged IMAC column (Cytiva). Contaminants were removed by washing with lysis buffer supplemented with 40 mM imidazole, and PARN was eluted in lysis buffer supplemented with 250 mM imidazole. PARN was further purified by size exclusion chromatography on a Superdex 200 26/600 column (Cytiva) in a buffer containing 10 mM HEPES/NaOH pH 7.5, 200 mM NaCl, 5% (v/v) glycerol, 2 mM DTT. The peak fractions were then pooled, concentrated to 0.9 mg/ml, flash-frozen in liquid nitrogen, and stored at −80 °C.

### Production and reconstitution of ternary complexes for cryo-EM
For reconstitution of the NOT1:NOT7:PieF complex, all three proteins were purified in parallel, as described above, and then mixed after the nickel-affinity chromatography step. The proteins were left to incubate on ice overnight. The next day, the complex was loaded and eluted on Superdex 200 26/600 (Cytiva) equilibrated in a buffer containing 10 mM HEPES/NaOH pH 7.5, 200 mM NaCl, 2 mM DTT. Reconstitution of the NOT1:NOT8:PieF was done identically to the NOT7 version of the complex. NOT1:NOT7:PieF and NOT1:NOT8:PieF were concentrated to 6.5 mg/ml and 8.3 mg/ml, respectively, mixed with 0.7 mM fluorinated octyl maltoside (Anatrace), and adsorbed onto glow-discharged holey carbon-coated grids (Quantifoil 300 mesh, Au R1.2/1.3) for 10 s. Grids were then blotted for 2 s at 100% humidity at 10 °C and frozen in liquid ethane using a Vitrobot Mark IV (Thermo Fisher Scientific).

### Cryo-EM data collection, data processing, and model building
Both NOT1:NOT7:PieF and NOT1:NOT8:PieF datasets were collected in counted mode (in Electron Event Representation format) on a CFEG-equipped Titan Krios G4 (Thermo Fisher Scientific) operating at 300 kV and × 165,000 magnification. A Selectris X imaging filter (Thermo Fisher Scientific) with slit width set to 10 eV on a Falcon 4i direct detection camera (Thermo Fisher Scientific) with a calibrated pixel size of 0.732 Å was used. Movies were collected at a total dose of 55.6 e⁻/Å$^2$ (NOT1:NOT7:PieF) or 52.4 e⁻/Å$^2$ (NOT1:NOT8:PieF), both fractionated to ~1 e⁻/Å$^2$/fraction for motion correction.

Patched motion correction, CTF parameter estimation, particle picking, extraction, and initial 2D classification were performed in SIMPLE 3.0[39]. All downstream processing was done in cryoSPARC or RELION, using the csparc2star.py script in the UCSF pyem package to convert between formats[40–42]. Global resolution was estimated from gold-standard Fourier shell correlations (FSCs) using the 0.143 criterion, and local resolution estimation was calculated within cryoSPARC using an FSC threshold of 0.5.

The cryo-EM processing workflow for NOT1:NOT7:PieF is outlined in Supplementary Fig. 12. Briefly, particles were subjected to two

rounds of reference-free 2D classification ($k = 300$) using a 140 Å soft circular mask. Four volumes were generated from a 225,824 particle subset of the 2D-cleaned particles after multi-class ab initio reconstruction using a maximum resolution cutoff of 7 Å. Output volumes were low-pass-filtered to 7 Å and used as references for a 4-class heterogeneous refinement against the full 2D-cleaned particle set (1,062,166 particles). Particles (417,581) from the most populated and structured class were selected and non-uniform refined against their corresponding volume lowpass-filtered to 15 Å, generating a 3.0 Å map. Bayesian polishing was performed in RELION followed by non-uniform refinement against the pre-polished volume, lowpass-filtered to 15 Å, yielding a 2.9 Å volume that was further improved to 2.8 Å after local and global CTF refinement (fitting tilt and trefoil).

The cryo-EM processing workflow for NOT1:NOT8:PieF is outlined in Supplementary Fig. 13. Briefly, particles were subjected to two rounds of reference-free 2D classification ($k = 200$) using a 150 Å soft circular mask. Selected particles (1,384,683) were then subjected to heterogeneous refinement against the four ab initio volumes generated from the NOT1:NOT7:PieF dataset, low-pass filtered to 7 Å. Particles (407,968) from the most populated and structured class were selected and non-uniformly refined against their corresponding volume lowpass-filtered to 15 Å, generating a 3.5 Å map. Bayesian polishing followed by 2D classification ($k = 100$, 150 Å soft circular mask) resulted in the selection of 200,675 particles. These particles were non-uniform refined against their corresponding pre-polished volume, low-pass-filtered to 15 Å, yielding a 3.5 Å volume that could be marginally improved by subsequent local CTF refinement. The "tightTarget" model of deepEMhancer was used on the final NOT1:NOT8:PieF half maps for postprocessing to improve map connectivity and side chain density[43].

Atomic models for NOT1:NOT7:PieF and NOT1:NOT8:PieF were generated first by docking AlphaFold2-Multimer models into their respective cryo-EM volumes, followed by manual rebuilding and real-space refinement in Coot v0.9.8.3[44,45]. Both models were further refined in real space using PHENIX with rotamer, Ramachandran restraints, and with or without the secondary structure restraints (the latter applied only for NOT1:NOT8:PieF) against either the global B-factor sharpened map of NOT1:NOT7:PieF or the deepEMhancer map for NOT1:NOT8:PieF, yielding the models described in Supplementary Table 2[43]. All models were validated using MolProbity within PHENIX[46,47].

## Analytical size exclusion chromatography

Individual proteins His$_6$-SUMO-NOT7 (10000 pmol), His$_6$-NOT1 (E1093–S1317) (15000 pmol), His$_6$-PieF (15000 pmol) and complexes His$_6$-SUMO-NOT7/His$_6$-NOT1(E1093–S1317), His$_6$-SUMO-NOT7/His$_6$-PieF, His$_6$-SUMO-NOT7/His$_6$-NOT1(E1093–S1317)/His$_6$-PieF (mixed at the same ratio to His$_6$-SUMO-NOT7) were loaded on Superdex 200 10/300 (Cytiva) equilibrated buffer containing 10 mM HEPES/NaOH pH 7.5, 200 mM NaCl, 2 mM DTT. Complexes were incubated on ice for 30 min before loading on the column. Detection of the complexes and individual proteins was tracked using UV absorbance at 280 nm. The content of the peak fractions was analyzed using SDS-PAGE electrophoresis followed by Coomassie staining. Chromatograms were exported and re-plotted in one graph using Prism10.

## Mass photometry

Mass photometry of His$_6$-NOT1(E1093–S1317):His$_6$-SUMO-NOT7:His$_6$-PieF was performed using the Refeyn TwoMP mass photometry instrument in a buffer containing 10 mM HEPES/NaOH pH 7.5, 200 mM NaCl, 2 mM DTT. Molecular weight calibrations were performed using two protein oligomer solutions, β-amylase (56, 112, and 224 kDa) and Thyroglobulin (670 kDa). The data acquisition was performed with AcquireMP (v2023 R1.1) software and data analysis using DiscoverMP (v2023 R1.2) software.

## Deadenylation assays

Deadenylation reactions were carried out at 37 °C in a buffer containing 20 mM PIPES/NaOH pH 7.0, 40 mM NaCl, 10 mM KCl and 2 mM Mg(OAc)$_2$. A purified His$_6$-SUMO-NOT7 (500 nM), His$_6$-SUMO-NOT8 (500 nM), or NOT6:NOT7 heterodimer (250 nM) was mixed with the PieF variant at the equimolar ratio and left to incubate on ice for 15 min. In the assays with PARN (2 nM), PieF was added in 20-fold excess. To start the reaction, synthetic 5′-fluorescein-labeled UCUACAU-A$_{20}$ substrate (50 nM) was added. To stop the reaction at the corresponding time point, 3x reaction volumes of RNA loading dye were added (95% [v/v] deionized formamide, 17.5 mM EDTA pH 8, 0.01% [w/v] bromophenol blue). The products were resolved on a denaturing TBE-urea polyacrylamide gel, which was subsequently imaged using Sapphire FL Biomolecular Imager (Azure Biosystems) or Amersham Typhoon Biomolecular Imager (Cytiva). Poly(A) tail length changes were quantified by plotting the most abundant tail length at each time point[48]. The apparent deadenylation rate (As/min) was estimated using linear regression.

## Isothermal titration calorimetry

ITC measurements were performed using an iTC200 calorimeter (Malvern Panalytical/MicroCal, Netherlands/USA) at 25 °C. Recombinant purified protein samples were dialyzed against 10 mM HEPES pH 7.5, 200 mM NaCl buffer at 4 °C, and final concentrations determined by UV absorbance at 280 nm. Each experiment consisted of 18 injections of 2.1 μl of 200 μM His$_6$-tagged PieF variant into the cell containing 20 μM His$_6$-SUMO-tagged NOT7 or NOT8 made 180 s apart. These were preceded by an initial 0.5 μl injection to accommodate the reagent's interaction during pre-titration thermal equilibration at the tip of the syringe. The initial injection was removed during data processing. Reference titrations were subtracted from experimental data, and thermodynamic parameters were determined using a one-site binding model in Origin 7.0 Malvern/MicroCal data analysis software. All ITC run parameters are summarized in Supplementary Table 1. Representative thermograms are shown in Fig. 1, Supplementary Figs. 1 and 6.

## Streptavidin pull-down assays

StrepII-tagged (C-terminally) MBP, StrepII-tagged (N-terminally) PieF, StrepII-tagged (C-terminally) and MBP-tagged (N-terminally) RNF219 (residues S434–Q600), StrepII-tagged (C-terminally) His$_6$-SUMO-tagged (N-terminally) NOT1 (E1093–S1317) were produced in E. coli BL21 (DE3) Star cells (Thermo Fisher Scientific) grown in autoinduction medium overnight at 37 °C. Cells were resuspended in binding buffer (50 mM HEPES/NaOH pH 7.5, 200 mM NaCl) and lysed by sonication. Lysate was cleared by centrifugation at 40,000 × g for 1 h at 4 °C. The cleared lysates were incubated with StrepTactin Sepharose resin (Cytiva) for 1 h at 4 °C. After the incubation, beads were washed three times with wash buffer (50 mM HEPES/NaOH pH 7.5, 200 mM NaCl, 0.03% [v/v] Tween) and once with binding buffer. In pull-down from Fig. 1b, 2000 pmol of purified His$_6$-SUMO-NOT7, His$_6$-SUMO-NOT8, and NOT9 module were added to bead-bound proteins and incubated for 1 h at 4 °C. In Fig. 1c, 500 pmol of His$_6$-SUMO-NOT7, NOT6:NOT7 and NOT6L:NOT7 were added. In the competition pull-down assay in Supplementary Fig. 1g, 500 pmol of His6-NOT6:StrepII-SUMO-NOT7 were immobilized, and 1–200 fold molar excess of His$_6$-PieF was added. In the pull-down assay in Supplementary Fig. 4b, 500 pmol of MBP-Strep, His$_6$-StrepII-SUMO-NOT7, His$_6$-NOT6:StrepII-SUMO-NOT7, NOT6$_{Δ1-28}$:StrepII-NOT7 were immobilized and 100-fold molar excess of PieF was added. In the competition pull-down assay in Supplementary Fig. 4c, 500 pmol of NOT6$_{Δ1-28}$:StrepII-NOT7 were immobilized, and 1–200 fold molar excess of PieF was added. After the incubation, bound proteins were washed three times with the binding buffer. Beads were incubated for 1 h at 4 °C in 40 μl of binding buffer supplemented with 50 mM biotin to elute the complexes. The eluted proteins were analyzed using SDS-polyacrylamide gel electrophoresis followed by Coomassie blue staining.

## Transfection, flow cytometry, and total RNA isolation

HEK293T cells were maintained in Dulbecco modified Eagle medium (DMEM) (Gibco) containing 10% (v/v) fetal bovine serum (FBS) and 1× GlutaMAX (Gibco), at 37 °C with 5% $CO_2$. HEK293T cells were seeded at a density of $0.5 \times 10^6$ cells per well in six-well plates. The next day, cells were transfected with 10 µg of plasmid in a complex with 10 µl lipofectamine 2000 (Invitrogen); after 16 h the media was replaced. Cells were reseeded on day 2 and day 4 and counted on day 2, day 4, and day 6 after transfection. For cell cycle analysis, cells were resuspended in PBS and stained with fixable Viability Dye eFluor 780 (ThermoFisher) before fixation and permeabilization using the manufacturer's instructions (BD Biosciences), followed by staining with DAPI and analysis by flow cytometry. Data were analyzed with FlowJo v10.9.0 software. Total RNA was extracted from one well of a 6-well plate on day 4 after transfection using TRI Reagent (Zymo) and Direct-zol RNA Microprep kit (Zymo) following the manufacturer's instructions.

## Nanopore sequencing and data analysis

Libraries for direct RNA sequencing using library kit SQK-RNA004 (Oxford Nanopore Technologies) and sequenced with PromethION (Oxford Nanopore Technologies) using FLO-PRO004RA flow cells. One flow cell was used for each sample. Basecalling and poly(A) tail length estimates were performed using Dorado v0.0.7 (Oxford Nanopore Technologies) on a GPU-enabled computing cluster. The violin plots of mRNA poly(A) tail length distributions were computed using the ggplot2 package in R v4.3.2.

## Resource availability

Further information and requests for resources and reagents should be directed to and will be fulfilled by E.V., subject to a completed materials transfer agreement.

## Reporting summary

Further information on research design is available in the Nature Portfolio Reporting Summary linked to this article.

# Data availability

The data supporting the findings of this study are available from the corresponding authors upon request. The generated cryo-EM maps and PDB codes associated with different structures are deposited in the EMDB and PDB databases under accession codes 47689 / 47690 and 9E7T / 9E7U, respectively. The nanopore direct RNA sequencing data are deposited in the NCBI Sequence Read Archive (SRA) database under accession code PRJNA1263024. All raw data associated with the gel images and deadenylation assays is provided in Source Data file.

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

## Acknowledgements
We are grateful to Tom Misteli, Bobby Hogg, and Robin Stanley for critical comments on the manuscript. We also thank members of the Valkov and Lea laboratories for helpful discussions, as well as our colleagues at the RNA Biology Laboratory and the Center for Structural Biology for their support and advice. This study was supported by the Intramural Research Program of the National Institutes of Health. F.P. was supported by a Walter Benjamin postdoctoral fellowship from the German Research Foundation (Deutsche Forschungsgemeinschaft) [Project number 531520533].

## Author contributions
Conceptualization: E.V. Methodology: Y.L., E.V., J.C.D., D.J.T., C.T.P. Investigation: Y.L., C.T.P., D.J.T., J.C.D., F.P., S.G.T. Resources: F.P., A.L.V. Formal analysis: Y.L., J.C.D., D.J.T., C.T.P., S.G.T., S.M.L., E.V. Validation: Y.L., C.T.P., A.L.V. Visualization: Y.L., J.D., D.J.T., C.T.P., S.M.L., E.V. Data curation: E.V., J.C.D., Y.L. Project administration: E.V. Supervision: E.V., S.M.L. Software: E.V. Writing—original draft: Y.L., J.C.D., D.J.T., E.V., S.M.L. Writing—review and editing: E.V., Y.L., S.M.L., C.T.P., J.C.D.

## Funding

## Competing interests
The authors declare no competing interests.
