## [Transparent Peer Review file · Nature Communications]

Intracellular Pathogen Effector Reprograms Host Gene Expression By Inhibiting mRNA Decay

Corresponding Author: Dr Eugene Valkov

Version 0:

Reviewer comments:

Reviewer #1

(Remarks to the Author)
NCOMMS-24-78425-T

Levdansky et al investigate the molecular basis for CNOT7/8 inhibition by the Legionella effector PieF through biochemical reconstitution and high-resolution cryo-EM structure determination of complexes between a fragment of CNOT1 in complex with CNOT7/PieF or CNOT8/PieF in a manuscript entitled 'Intracellular Pathogen Effector Reprograms Host Gene Expression By Inhibiting mRNA Decay'.

The Legionella effector PieF was identified as a CCR4-NOT interactor in two papers published in bioRxiv in 2022. In this manuscript, the authors show that PieF specifically interacts with CNOT7 or CNOT8, that interactions with PieF are mutually exclusive with CNOT7/8 interactions with CNOT6, and that PieF is a potent inhibitor of deadenylase activity of CNOT7/8, but not of CNOT6/CNOT7/8. They also show that overexpression of PieF in cells leads to a substantial expansion of polyA tail lengths as well contributing to cell cycle defects. While this is true, the authors acknowledge that PieF levels are unlikely to reach comparable levels during the normal course of infection and that PieF deletion does not inhibit Legionella infection at a gross level. With that said, given the high affinity of PieF for CNOT7/8, and that CNOT6 is substochiometric to CNOT7/8 and other components of CCR4-NOT, it is highly likely that PieF functions as a CNOT7/8 inhibitor during infection, thus opening the door to understanding how it contributes to this process and pathology.

The high-resolution structures show that the PieF C-term tail reaches into the CNOT7/8 active site to occlude it, while also displacing a magnesium ion that is critical for CNOT7/8 catalytic activity. They perform structure/function analysis to confirm the importance of critical residues involved in binding as well as interactions within the CNOT7/8 active site.

There are no experimental issues that warrant further attention, and there are no major issues with the manuscript as presented. All the structural work appears sound.

On minor point. The observation/reference that CNOT6 is substochiometric to CNOT7/8 is left to the discussion. It might help to cite that information just before or during presentation of data presented in Figure 1e. As it is currently presented, it leaves the reader wondering if PieF interactions with CNOT7/8 are relevant as most people incorrectly think of CNOT6 as a stoichiometric component of the CCR4-NOT complex.

Reviewer #2

(Remarks to the Author)

The authors present results of a study on the effector protein from the intracellular bacterial pathogen Legionella pneumophila, PieF, which was shown in earlier studies to interact with the eukaryotic CCR4-NOT deadenylase complex. The current study characterises the interaction and shows that it has specificity for two subunits of the complex and modulates the activity of the complex in vitro and in vivo. The work includes biophysical characterisation of the interactions and detailed structural analysis by cryoEM. The authors show that PieF impedes the deadenylation activity of the CCR4-NOT complex not other DEDD-class of exoribonucleases. This is rationalised by cryoEM structures provided by the authors

of the NOT1:NOT7:PieF complex at 2.8 Å resolution and NOT1:NOT8:PieF complex to 3.5 Å resolution. PieF increases modal mRNA length when overexpressed and disrupts normal cell cycle progression in human tissue culture cells, and the inference is that this is through inhibition of deadenylation.

It is intriguing that, despite the strong equilibrium binding, PieF cannot displace NOT6 from the NOT6:NOT7 heterodimer, suggesting a kinetic bottleneck for access of the effector to the pre-assembled NOT sub-assembly. This raises the question of when PieF could access the NOT7 (or NOT8) and how effective an inhibitor it would be expected to be in vivo. Surprisingly, it still seems to work as expected in the human cells. The authors suggest that assembly of the NOT complex is a key step for normal function.

The implication of the work is that pathogen targets suppression of deadenylation
The experimental work is solid and the results are presented in a clear and compelling manner. There are a few minor comments that will hopefully be useful for the authors.

Cells used for the analysis were HEK293T, which are transformed cells. Is there a possibility that the PieF impact on cell cycle is complicated because these are transformed cells?

Line 18 abstract – tail shortening with selectivity – for target transcripts?

Line 26, please define cia-dependent

Line 61, would the following suggestion for subsection title be more accurate?

Legionella pneumophila PieF effector binds to specific CCR4-NOT subunits

Line 71 might read better as “NOT7 deadenylase is a component...” since it was already mentioned earlier that it is a DEDD-type enzyme.

Line 188: The title "PieF disrupts mRNA poly(A) tail control and triggers cell cycle arrest in human cells" could be changed to "PieF disrupts mRNA poly(A) tail control and impairs cell cycle progression in human cells"?
The findings strongly support that PieF inhibits the CCR4-NOT complex, which regulates mRNA deadenylation (i.e., shortening of the poly(A) tail). However, while the data suggests that PieF disrupts cell cycle progression (e.g., by decreasing G0/G1 cells and increasing S phase cells), this does not fully align with the concept of cell cycle arrest, which typically refers to the activation of cell cycle checkpoints halting progression at a specific phase. Instead, PieF appears to push cells into S phase, where they may become stalled or progress inefficiently (which would explain their lower proliferation rates). Evidence of a true arrest would require more markers (e.g., checkpoint activation, phosphorylated p53, or cyclin-dependent kinase inhibitors).

Line 256 might read better as "... is essential for successful replication of the bacterium."

Extended Figure 1f and extended Figure 6 c,d,f – the titration mid points for the ITC profiles seem closer to 0.7. Is there a systematic error in protein concentration estimates?

Extended figure 6b could incorporate the curves for “NOT7” and “NOT7 + PieF WT” to be able to compare the effects of the different mutations on PieF inhibitory activity in one Figure.

Reviewer #3

(Remarks to the Author)

Levdansky et al. present a well-written and highly focussed analysis of the structural and functional consequences of the Legionella pneumophila PieF effector protein as it enters eukaryotic cells via a pathogen secretion system. By combining in vitro protein-protein interaction and RNA decay studies with structural biology and, eventually, measurements of poly-A tail lengths in HeLa cells in vivo, the authors carefully delineate the mechanism by which the PieF protein binds and inactivates the host CCR4-NOT complex, leading to serious disturbances in mRNA decay and consequently, control of mRNA lifetime. In the end, this allows the pathogen to control the host cell cycle and, presumably, prolongue the period of infection before the innate immune system can intervene due to inhibition of expression of central transcription factors.

Overall, the manuscript is convincing and very well written and experiments well planned and executed, figures are (with a few exceptions, see below) very clear and follow a consistent colour scheme. I am particularly impressed by the visualisation of the structures on Fig. 2 that are exceptionally clear and true to experimental data (the EM map). The story is nicely focused on a few central take-home messages and provides clear, mechanistic evidence for what is likely a key immune evasion mechanism in Legionella, an important human pathogen. The paper is therefore likely to be of interest to a broad readership and will serve as a cornerstone in our understanding of this pathogen in the future. I have a few minor concerns and suggestions, which are listed below, but apart from that, I believe this manuscript should be published with minor revisions.

Minor comments:

Line 113. Cryo-EM section, page 4. If the fold of PieF is new, which is a little bit surprising given that it looks like a relatively standard beta-sheet with surrounding helices, I suggest you spend a few words describing the fold for future reference.

Line 117-124. The mechanism by which PieF interferes and interacts with NOT7 whereby it displaces a catalytic Mg²⁺ ion is reminiscent of some bacterial toxin-antitoxin complexes in which the toxin is of the RNA interferase type and where the antitoxin also uses a positively charged side chain to displace a divalent metal ion. I wonder if this comparison could yield any additional insights?

Line 134-138 and Figure 3. It looks to me as if PieF is really a structural NOT6 mimic (apart from the extension in NOT6). Would it be worth spending a bit of time on discussing this from an evolutionary point of view? Did PieF arise independently in the bacterial lineage and presents a case of convergent evolution or could Legionella (to which this protein appears unique) somehow have acquired a copy of the mammalian gene at some point? I guess sequence comparison should be able to tell if there is reason to believe this.

Line 143. The conclusion that the additional extension or loop in NOT6 is what prevents PieF from disrupting the NOT6:NOT7 complex is lacking experimental support. Could you challenge this idea by removing the extension and checking for displacement in vitro? It looks like one could relatively easily design a shorter loop version of NOT6 without disrupting the main fold.

Line 154-5. Could you speculate (perhaps in the Discussion) on the implications of the observation that PieF does not prevent incorporation of the inhibited deadenylase into the CCR4-NOT complex?

Line 70. This sentence is repetitive, right?

Line 101. "and" missing between "domain" and "PieF" in the beginning of the line.

Figure 3 + Extended Data Fig. 3 and 4. I am missing distance indications on central interactions. Also, I am not convinced that all interactions are H bonds. What were the criteria for showing interactions (distance + angle)? For example, in Ext. Data Fig. 3b, the two interactions between R49 and E117 cross in a strange way.

Figure 3 + Extended Data Fig. 3 and 4. The blue colour used for the side chain sticks makes it very hard to discern C from N. Consider brightening the carbon colour.

Version 1:

Reviewer comments:

Reviewer #2

(Remarks to the Author)

The authors have provided compelling replies to all the reviewer comments and have improved the manuscript. All the changes to the revised manuscript are satisfactory. One minor point is that it would be helpful to annotate Figure 3c,d to show the Not6 M1-K28 region that is referred to on line 153.

Reviewer #3

(Remarks to the Author)

The authors have appropriately addressed my concerns and I have no further comments or suggestions. Nice work!

AUTHOR RESPONSE TO REVIEWERS

We thank the reviewers for their enthusiastic, positive, and thoughtful feedback. Despite ongoing, unprecedented disruptions to our research operations, we have constructively and diligently addressed every comment by all reviewers and performed additional experiments using an N-terminally truncated NOT6. Our response to the comments is in blue font, and changes to the manuscript text are in dark red.

Reviewer #1 (Remarks to the Author):

Levdansky et al investigate the molecular basis for CNOT7/8 inhibition by the Legionella effector PieF through biochemical reconstitution and high-resolution cryo-EM structure determination of complexes between a fragment of CNOT1 in complex with CNOT7/PieF or CNOT8/PieF in a manuscript entitled 'Intracellular Pathogen Effector Reprograms Host Gene Expression By Inhibiting mRNA Decay'.

The Legionella effector PieF was identified as a CCR4-NOT interactor in two papers published in bioRxiv in 2022. In this manuscript, the authors show that PieF specifically interacts with CNOT7 or CNOT8, that interactions with PieF are mutually exclusive with CNOT7/8 interactions with CNOT6, and that PieF is a potent inhibitor of deadenylase activity of CNOT7/8, but not of CNOT6/CNOT7/8. They also show that overexpression of PieF in cells leads to a substantial expansion of polyA tail lengths as well contributing to cell cycle defects. While this is true, the authors acknowledge that PieF levels are unlikely to reach comparable levels during the normal course of infection and that PieF deletion does not inhibit Legionella infection at a gross level. With that said, given the high affinity of PieF for CNOT7/8, and that CNOT6 is substochiometric to CNOT7/8 and other components of CCR4-NOT, it is highly likely that PieF functions as a CNOT7/8 inhibitor during infection, thus opening the door to understanding how it contributes to this process and pathology.

The high-resolution structures show that the PieF C-term tail reaches into the CNOT7/8 active site to occlude it, while also displacing a magnesium ion that is critical for CNOT7/8 catalytic activity. They perform structure/function analysis to confirm the importance of critical residues involved in binding as well as interactions within the CNOT7/8 active site.

There are no experimental issues that warrant further attention, and there are no major issues with the manuscript as presented. All the structural work appears sound.

Thank you for these kind remarks.

On minor point. The observation/reference that CNOT6 is substochiometric to CNOT7/8 is left to the discussion. It might help to cite that information just before or during presentation of data presented in Figure 1e. As it is currently presented, it leaves the reader wondering if PieF

interactions with CNOT7/8 are relevant as most people incorrectly think of CNOT6 as a stoichiometric component of the CCR4-NOT complex.

Yes, indeed. This is a great point and one we have pondered extensively. Having taken a deeper dive into the older literature, there is additional support for this observation from Lau et al. (PMID: 19558367), where the authors carried out a systematic quantitative analysis of the ratios of CCR4-NOT subunits from human cells using mass spectrometry. Here, they also observed that NOT6 is strikingly substoichiometric to other CCR4-NOT subunits. We have added a couple of sentences to the results leading up to Fig. 1e to introduce this concept of bias in CCR4-NOT composition in a more physiological setting.

Reviewer #2 (Remarks to the Author):

The authors present results of a study on the effector protein from the intracellular bacterial pathogen *Legionella pneumophila*, PieF, which was shown in earlier studies to interact with the eukaryotic CCR4-NOT deadenylase complex. The current study characterises the interaction and shows that it has specificity for two subunits of the complex and modulates the activity of the complex in vitro and in vivo. The work includes biophysical characterisation of the interactions and detailed structural analysis by cryoEM. The authors show that PieF impedes the deadenylation activity of the CCR4-NOT complex not other DEDD-class of exoribonucleases. This is rationalised by cryoEM structures provided by the authors of the NOT1:NOT7:PieF complex at 2.8 Å resolution and NOT1:NOT8:PieF complex to 3.5 Å resolution. PieF increases modal mRNA length when overexpressed and disrupts normal cell cycle progression in human tissue culture cells, and the inference is that this is through inhibition of deadenylation.

It is intriguing that, despite the strong equilibrium binding, PieF cannot displace NOT6 from the NOT6:NOT7 heterodimer, suggesting a kinetic bottleneck for access of the effector to the pre-assembled NOT sub-assembly. This raises the question of when PieF could access the NOT7 (or NOT8) and how effective an inhibitor it would be expected to be in vivo. Surprisingly, it still seems to work as expected in the human cells. The authors suggest that assembly of the NOT complex is a key step for normal function.

The implication of the work is that pathogen targets suppression of deadenylation
The experimental work is solid and the results are presented in a clear and compelling manner. There are a few minor comments that will hopefully be useful for the authors.

Thank you for your comments and diligence in helping us improve this study.

Cells used for the analysis were HEK293T, which are transformed cells. Is there a possibility that the PieF impact on cell cycle is complicated because these are transformed cells?

The reviewer raises a valid point regarding using HEK293T cells, which are indeed transformed and thus differ from non-transformed or primary cell lines in aspects of cell cycle regulation. We acknowledge that this transformation could potentially influence how PieF impacts the cell cycle. The only way to address this, however, is to carry out future experiments utilizing non-transformed or primary cell lines to confirm and generalize our findings. Ideally, one would assess PieF's effects on *Legionella* intracellular survival and host- cell cycle progression in primary macrophages from permissive and restrictive mouse strains using PieF- deficient and wt bacteria; however, this lies beyond our current scope. Nonetheless, overexpressing PieF in HEK293T cells, a widely used, tractable model for mechanistic cell- cycle studies, can provide valuable initial insights to stimulate future work.

Line 18 abstract – tail shortening with selectivity – for target transcripts?

Yes, this statement could be misinterpreted in this way. We revised to clarify that PieF is selective for NOT7/8 enzymes in CCR4-NOT, unlike other DEED-type ribonucleases.

Line 26, please define cia-dendendent

This is a group of small RNAs that are controlled by the two-component regulatory system CiaRH, and are widely conserved in streptococci. We added this clarification to the text and cited the relevant work.

Line 61, would the following suggestion for subsection title be more accurate?

Legionella pneumophila PieF effector binds to specific CCR4-NOT subunits

It would indeed! Thank you for this and your other thoughtful suggestions. They are appreciated.

Line 71 might read better as “NOT7 deadenylase is a component...” since it was already mentioned earlier that it is a DEED-type enzyme.

Done.

Line 188: The title "PieF disrupts mRNA poly(A) tail control and triggers cell cycle arrest in human cells" could be changed to "PieF disrupts mRNA poly(A) tail control and impairs cell cycle progression in human cells"?

Done. Thank you for the suggestion.

The findings strongly support that PieF inhibits the CCR4-NOT complex, which regulates mRNA deadenylation (i.e., shortening of the poly(A) tail). However, while the data suggests that PieF disrupts cell cycle progression (e.g., by decreasing G0/G1 cells and increasing S

phase cells), this does not fully align with the concept of cell cycle arrest, which typically refers to the activation of cell cycle checkpoints halting progression at a specific phase. Instead, PieF appears to push cells into S phase, where they may become stalled or progress inefficiently (which would explain their lower proliferation rates). Evidence of a true arrest would require more markers (e.g., checkpoint activation, phosphorylated p53, or cyclin-dependent kinase inhibitors).

Thank you for pointing out the nuanced difference between cell cycle “arrest” and impaired progression. We agree that our current data indicate that PieF overexpression drives cells into S phase and slows or disrupts cell cycle progression rather than causing a classical checkpoint-mediated arrest. Specifically, our observations of decreasing G0/G1 and increasing S phase populations suggest that cells progress into S phase but do so inefficiently, leading to reduced proliferation. We revised the section title to “PieF disrupts mRNA poly(A) tail control and impairs cell cycle progression in human cells,” which indeed better captures our findings. We have also updated the text in the Results and Discussion sections to clarify the distinction between a checkpoint-mediated arrest and impaired progression.

Line 256 might read better as “... is essential for successful replication of the bacterium.”

Done.

Extended Figure 1f and extended Figure 6 c,d,f – the titration mid points for the ITC profiles seem closer to 0.7. Is there a systematic error in protein concentration estimates?

Thank you for your insightful comment regarding the apparent midpoints (~0.7) observed in our ITC titrations and the potential concern about protein concentration estimates. We have carefully investigated this possibility.

To estimate protein concentrations, we used UV absorbance at 280 nm, with the extinction coefficient calculated from the known amino acid sequence. To ensure accuracy, we performed measurements on at least two independent protein preparations. The results were consistent, indicating that our initial protein concentration estimates are accurate within the typical 5–10% experimental variation.

Multiple ITC experiments were performed using freshly prepared protein samples. The reproducibility of the thermodynamic parameters (within standard experimental error) further suggests that the protein concentration is not systematically over- or underestimated.

Even when protein concentrations are accurate, several common experimental or biological factors can lead to an apparent stoichiometry near 0.7 rather than 1.0. A fraction of the protein could be misfolded or aggregated, rendering some fraction of the binding sites unavailable. Alternatively, if the protein exists in more than one conformational state in solution, where only one of the states binds ligand efficiently, the overall apparent stoichiometry can be reduced.

Occasionally, subtle deviations in baseline subtraction or the curve-fitting routine produce stoichiometries slightly below 1.0.

We performed ITC with different protein-to-titrant starting ratios and consistently obtained comparable stoichiometries, suggesting that the shift is not solely due to errors in our stock concentrations. We analyzed the protein using SDS-PAGE, size-exclusion chromatography, and mass photometry before ITC to confirm purity and oligomeric state. No significant aggregates or impurities were detected. We critically re-examined our ITC raw data and curve fittings and consistently observed the same midpoint shift.

Given these extensive checks, we have confidence that the protein concentration estimates are accurate within typical experimental errors. We hypothesize that the observed midpoint shift is more likely due to a small fraction of non-binding protein (or another form of heterogeneity) rather than a systematic concentration error. Indeed, slight deviations from an exact stoichiometry of 1.0 are not uncommon in complex protein-ligand systems.

We appreciate the reviewer's attention to this detail and hope we have clarified the possible reasons for the observed ITC profiles.

Extended figure 6b could incorporate the curves for "NOT7" and "NOT7 + PieF WT" to be able to compare the effects of the different mutations on PieF inhibitory activity in one Figure.

Done.

Reviewer #3 (Remarks to the Author):

Levdansky et al. present a well-written and highly focussed analysis of the structural and functional consequences of the *Legionella pneumophila* PieF effector protein as it enters eukaryotic cells via a pathogen secretion system. By combining in vitro protein-protein interaction and RNA decay studies with structural biology and, eventually, measurements of poly-A tail lengths in HeLa cells in vivo, the authors carefully delineate the mechanism by which the PieF protein binds and inactivates the host CCR4-NOT complex, leading to serious disturbances in mRNA decay and consequently, control of mRNA lifetime. In the end, this allows the pathogen to control the host cell cycle and, presumably, prolongue the period of infection before the innate immune system can intervene due to inhibition of expression of central transcription factors.

Overall, the manuscript is convincing and very well written and experiments well planned and executed, figures are (with a few exceptions, see below) very clear and follow a consistent colour scheme. I am particularly impressed by the visualisation of the structures on Fig. 2 that are exceptionally clear and true to experimental data (the EM map). The story is nicely focused on a few central take-home messages and provides clear, mechanistic evidence for what is likely a key immune evasion mechanism in *Legionella*, an important human pathogen. The

paper is therefore likely to be of interest to a broad readership and will serve as a cornerstone in our understanding of this pathogen in the future. I have a few minor concerns and suggestions, which are listed below, but apart from that, I believe this manuscript should be published with minor revisions.

Thank you for your kind comments and appreciation of our work.

Minor comments:

Line 113. Cryo-EM section, page 4. If the fold of PieF is new, which is a little bit surprising given that it looks like a relatively standard beta-sheet with surrounding helices, I suggest you spend a few words describing the fold for future reference.

Yes, in the strictest sense, the fold is relatively standard. We did not observe any structural similarity to any known proteins using Foldseek, which is what we based this original claim on. The reviewer is correct, and we have rephrased.

Line 117-124. The mechanism by which PieF interferes and interacts with NOT7 whereby it displaces a catalytic Mg²⁺ ion is reminiscent of some bacterial toxin-antitoxin complexes in which the toxin is of the RNA interferase type and where the antitoxin also uses a positively charged side chain to displace a divalent metal ion. I wonder if this comparison could yield any additional insights?

Thank you for this great suggestion! Indeed, certain bacterial toxin-antitoxin systems use a positively charged residue of the antitoxin to eject a catalytic metal ion from the toxin's active site, thereby inhibiting the toxin's RNase (RNA interferase) activity. Notable examples include FitA-FitB from *Neisseria gonorrhoeae*. In this system, the FitA antitoxin inserts an arginine side chain (Arg68) deep into the PIN-domain active site of the FitB toxin. This arginine not only blocks the active site but displaces the catalytic Mg²⁺ ion that FitB would usually require for RNA cleavage. As a result, the RNase activity of FitB is neutralized when complexed with FitA (PMID: 28508407). Another example is VapB-VapC from *Klebsiella pneumoniae*, where the VapB antitoxin inhibits its VapC toxin through an "Mg²⁺ switch" mechanism, in which a key arginine residue (R79 of VapB) occupies the toxin's active-site pocket and displaces the bound Mg²⁺ cofactor (PMID: 33146528).

We added this text to the Discussion: *"Interestingly, the inhibitory strategy of PieF parallels that of certain bacterial toxin-antitoxin systems. In these complexes, the antitoxin protein neutralizes its RNase toxin by physically occluding the active site and displacing the catalytic Mg²⁺ ion with a positively charged side chain. For example, the FitA antitoxin inserts an Arg residue into the FitB toxin's active site to expel the Mg²⁺ required for catalysis (PMID: 28508407). In another example, the VapB antitoxin uses an Arg (R79) to eject the Mg²⁺ cofactor from the VapC toxin (PMID: 33146528). These parallels underscore a convergent inhibitory strategy among bacterial pathogens, whereby PieF, similar to bacterial antitoxins, inactivates NOT7's enzymatic core by displacing its catalytic metal ion."*

Line 134-138 and Figure 3. It looks to me as if PieF is really a structural NOT6 mimic (apart from the extension in NOT6). Would it be worth spending a bit of time on discussing this from an evolutionary point of view? Did PieF arise independently in the bacterial lineage and presents a case of convergent evolution or could *Legionella* (to which this protein appears unique) somehow have acquired a copy of the mammalian gene at some point? I guess sequence comparison should be able to tell if there is reason to believe this.

Yes, we considered this possibility. It is outside of our expertise and the scope of this work to give a definitive assessment on whether PieF evolved independently or *Legionella* acquired the mammalian gene. We did not observe any sequence similarity between the two proteins, using the available tools.

Line 143. The conclusion that the additional extension or loop in NOT6 is what prevents PieF from disrupting the NOT6:NOT7 complex is lacking experimental support. Could you challenge this idea by removing the extension and checking for displacement in vitro? It looks like one could relatively easily design a shorter loop version of NOT6 without disrupting the main fold.

We appreciate the reviewers' suggestion and have accordingly engineered an N- terminal truncation of NOT6 to form the NOT6:NOT7 heterodimer. We evaluated this construct in competition pulldown and deadenylation assays (Supplementary Figure 4a–d). We found that PieF, even in significant excess, neither disrupts the NOT6:NOT7 association nor inhibits its enzymatic activity. These results demonstrate that the N- terminal extension of NOT6 does not confer resistance to PieF, providing a valuable additional insight into PieF's mechanism of action.

Line 154-5. Could you speculate (perhaps in the Discussion) on the implications of the observation that PieF does not prevent incorporation of the inhibited deadenylase into the CCR4-NOT complex?

We emphasized that PieF potentially traps the inhibited NOT7 within the assembled CCR4-NOT complex, rendering it catalytically incompetent.

Line 70. This sentence is repetitive, right?

Yes. This was altered.

Line 101. "and" missing between "domain" and "PieF" in the beginning of the line.

Done.

Figure 3 + Extended Data Fig. 3 and 4. I am missing distance indications on central interactions. Also, I am not convinced that all interactions are H bonds. What were the criteria

for showing interactions (distance + angle)? For example, in Ext. Data Fig. 3b, the two interactions between R49 and E117 cross in a strange way.

We based our analysis on the criteria implemented in the PDBe PISA (Proteins, Interfaces, Structures and Assemblies) tool (PMID: 17681537). This information is provided in the new Supplementary Tables 4 and 5. We fixed the representation in Supplementary Figure 3b. Thank you for pointing that out.

Figure 3 + Extended Data Fig. 3 and 4. The blue colour used for the side chain sticks makes it very hard to discern C from N. Consider brightening the carbon colour.

Done for Supplementary Figures. We felt that the clarity of Figure 3 would be compromised by using lighter colors for the side chains, given the transparency of the overlays, so it was left as is.

RESPONSE TO REVIEWERS

Reviewer #2 (Remarks to the Author):

The authors have provided compelling replies to all the reviewer comments and have improved the manuscript. All the changes to the revised manuscript are satisfactory. One minor point is that it would be helpful to annotate Figure 3c,d to show the Not6 M1-K28 region that is referred to on line 153.

We thank the reviewer for the suggestion. To show the NOT6 with the indicated missing N-terminal part (M1-K28), we made a separate supplementary figure panel (Supplementary Figure 4a, the missing part is highlighted in white) and added a reference to this panel in the text (line 153).

Reviewer #3 (Remarks to the Author):

The authors have appropriately addressed my concerns and I have no further comments or suggestions. Nice work!

Thank you!